# Clinical Manifestations, Macrolide Resistance, and Treatment Utilization Trends of *Mycoplasma pneumoniae* Pneumonia in Children and Adolescents in South Korea

**DOI:** 10.3390/microorganisms12091806

**Published:** 2024-08-31

**Authors:** Joon Kee Lee, Taekjin Lee, Yae-Jean Kim, Doo Ri Kim, Areum Shin, Hyun Mi Kang, Ye Ji Kim, Dong Hyun Kim, Byung Wook Eun, Young June Choe, Hyunju Lee, Young Min Cho, Eun Young Cho, Kyung Min Kim, Byung Ok Kwak, Su Eun Park, Kyo Jin Jo, Jae Hong Choi, Dayun Kang, Eun Hwa Choi, Ki Wook Yun

**Affiliations:** 1Department of Pediatrics, Chungbuk National University Hospital, Chungbuk National University College of Medicine, Cheongju 28644, Republic of Korea; leejoonkee@chungbuk.ac.kr; 2Department of Pediatrics, CHA Bundang Medical Center, CHA University, Seongnam 13496, Republic of Korea; bjloveu@daum.net; 3Department of Pediatrics, Samsung Medical Center, School of Medicine, Sungkyungwan University, Seoul 06351, Republic of Korea; yaejeankim@skku.edu (Y.-J.K.); doori756.kim@samsung.com (D.R.K.); areum90.shin@samsung.com (A.S.); 4Department of Pediatrics, College of Medicine, The Catholic University of Korea, Seoul 06591, Republic of Korea; pedhmk@gmail.com (H.M.K.); jenniferyejikim@gmail.com (Y.J.K.); 5Department of Pediatrics, Inha University College of Medicine, Incheon 22212, Republic of Korea; leicakim@gmail.com; 6Department of Pediatrics, Eulji University Eulji General Hospital, Seoul 01830, Republic of Korea; acet0125@hanmail.net; 7Department of Pediatrics, Korea University Anam Hospital, and Allergy and Immunology Center, Korea University, Seoul 02841, Republic of Korea; choey@korea.ac.kr; 8Department of Pediatrics, Seoul National University Bundang Hospital, Seongnam 13620, Republic of Korea; hyunjulee@snu.ac.kr (H.L.); rclahm@gmail.com (Y.M.C.); 9Department of Pediatrics, Seoul National University College of Medicine, Seoul 03080, Republic of Korea; eunchoi@snu.ac.kr; 10Department of Pediatrics, Chungnam National University Hospital, Daejeon 35015, Republic of Korea; pedeyc@gmail.com (E.Y.C.); kmkimpid@gmail.com (K.M.K.); 11Department of Pediatrics, Hallym University Kangnam Sacred Heart Hospital, Seoul 07441, Republic of Korea; qquack00@daum.net; 12Department of Pediatrics, Pusan National University Yangsan Hospital, Pusan National University School of Medicine, Yangsan 50612, Republic of Korea; psepse@naver.com (S.E.P.); godjkj@nate.com (K.J.J.); 13Department of Pediatrics, Jeju National University School of Medicine, Jeju 63243, Republic of Korea; pedidongs@gmail.com; 14Department of Pediatrics, Seoul National University Children’s Hospital, Seoul 03080, Republic of Korea; dayun0333@gmail.com

**Keywords:** child, adolescent, mycoplasma pneumoniae, macrolides

## Abstract

A resurgence of *Mycoplasma pneumoniae* (MP)—the leading cause of community-acquired bacterial pneumonia, particularly in children—occurred following the COVID-19 pandemic. We aimed to investigate the clinical manifestations, macrolide resistance patterns, and therapeutic approaches related to the MP pneumonia epidemic. Children and adolescents diagnosed with MP pneumonia in September–December 2023 were screened. Clinical data were retrospectively collected from 13 major hospitals using concordant microbiological criteria, including either a positive PCR result or four-fold increase in serological markers. Demographic characteristics, treatment modalities, and clinical outcomes were analyzed. Of the 474 screened patients, 374 (median age: 7.7 [IQR, 5.4–9.6] years; hospitalization rate: 88.6%) met the microbiological confirmation criteria. Most patients experienced fever (98.9%), and lobular/lobar consolidation (59.1%) was the dominant radiological finding. The macrolide resistance rate remained high at 87.0%; corticosteroids were widely used (55.6%) alongside macrolides, despite resistance. Patients with consolidation had prolonged fever (median 8 vs. 7 days, *p* = 0.020) and higher hospitalization rates (92.3% vs. 83.0%, *p* = 0.008). Macrolide resistance did not significantly influence radiological outcomes. This study highlights the ongoing challenge of macrolide resistance in MP pneumonia and need for tailored therapeutic approaches. Despite high resistance, macrolides remain commonly prescribed, often concurrently with corticosteroids.

## 1. Introduction

*Mycoplasma pneumoniae* (MP) has emerged as the predominant pathogen in cases of community-acquired pneumonia with bacterial origin, particularly in the post-pneumococcal conjugate vaccine era [1]. Due to its lack of a cell wall, beta-lactam antibiotics are ineffective against MP [2]. Consequently, antibiotics, such as macrolides, fluoroquinolones, and tetracyclines, have been utilized for the treatment of MP-associated diseases [3]. Since the discovery of macrolide-resistant MP (MRMP), the clinical efficacy of macrolides has been questioned, particularly in patients infected with MRMP or those who do not respond to macrolides when resistance is not tested [4]. Recent studies from Asian countries have shown a marked increase in macrolide resistance, although the exact reasons for this trend are not fully understood [5,6,7]. For instance, in Korea, the macrolide resistance rate increased from 0% in 2000 to 84.4% during the 2014–2016 epidemic years [8]. Therefore, the use of alternative medications is considered in severe cases or when there is no response to macrolides [9]. However, concerns have been raised regarding the safety of prescribing quinolones and tetracyclines to children and adolescents [3]. Thus, monitoring MP pneumonia epidemics and current treatment approaches is crucial for establishing optimal management strategies for MP infections.

Following the end of the COVID-19 pandemic, MP pneumonia epidemics have been documented in specific regions of China since June 2023, with a marked increase in incidence during the school season in September, particularly affecting children and adolescents [10,11,12]. Although the occurrence of an MP epidemic every 4–7 years for a duration of 2–3 years is not unprecedented, the significance of the most recent outbreak was heightened due to its timing following the cessation of non-pharmaceutical interventions (NPIs) [3,13,14]. Similarly, in South Korea, a resurgence of MP occurred in late 2023, following a previous epidemic in 2019–2020 [15,16]. Utilizing the nationwide monitoring system known as ARI Net, which detects respiratory pathogens in admitted patients at selected institutions, the Korea Disease Control and Prevention Agency announced a doubling in the number of admissions for MP management during the recent four weeks as of 17 November 2023 [17]. This was further confirmed by assessing the positive rates of polymerase chain reaction (PCR) tests conducted for MP in selected hospitals from this study (Appendix A).

This multi-center study aimed to systematically investigate the clinical manifestations, macrolide resistance patterns, and treatment utilization during the latest MP pneumonia epidemic among children and adolescents in South Korea.

## 2. Materials and Methods

### 2.1. Patient Selection and Data Collection

Children and adolescents up to 18 years old diagnosed with MP pneumonia between 1 September and 31 December 2023 were eligible for inclusion in the study. Clinical data were retrospectively collected from 13 academic university hospitals across South Korea. Initially, each hospital used its own microbiological diagnostic criteria for MP to review patients for radiologic evidence of pneumonia. The clinical data of patients with pneumonia and a microbiologic diagnosis of MP were then consolidated. Following consolidation, stringent uniform diagnostic criteria were adopted to exclude patients lacking sufficient evidence of MP diagnosis. The collected clinical data included demographic information (age and sex), clinical presentation (symptoms such as fever, cough, sputum, and dyspnea), vital signs, radiologic findings (chest X-ray and computed tomography results), laboratory results (white blood cell count, C-reactive protein levels, and liver function tests), microbiologic results (PCR and serology), treatment details (types and duration of antibiotics, corticosteroids, and other medications), hospitalization details (length of stay, need for intensive care, and mechanical ventilation), and clinical outcomes (recovery, complications, and in-hospital mortality).

### 2.2. Excluded Patients

Immunocompromised patients, those with chronic pulmonary diseases, and hospital-acquired infections were excluded from the study to ensure clarity in assessing clinical manifestations and prognosis specifically for community-acquired pneumonia. Exclusion criteria for immunocompromised status included long-term corticosteroid use (for conditions other than MP treatment), biologic therapy, primary immune deficiencies, hematologic malignancies, post-transplant status, chronic illnesses requiring immunosuppressive treatment, and the use of other immunosuppressive drugs. These exclusion criteria ensured that the study population was not influenced by altered immune responses or nosocomial factors, allowing for a more accurate assessment of MP pneumonia and its treatment outcomes.

### 2.3. Definition and Classification of Pneumonia

Pediatricians confirmed pneumonia diagnosis through the identification of chest radiographic infiltration along with clinical indicators including fever (defined as ≥38 °C), cough, and sputum. Radiographic findings were categorized into six groups: lobular/lobar consolidation, peribronchial/parahilar infiltration, patchy consolidation, nodules, total haziness (in one or both lungs), and ground-glass opacity (in one or both lungs). Additionally, the presence and extent of pleural effusion were assessed separately, with the effusion size classified as large or small based on a threshold of 1 cm. This study compared clinical manifestations and treatment outcomes across various radiological findings.

### 2.4. Microbiologic Criteria

Cases meeting any of the following criteria were included for further analysis: (1) MP detection by PCR in respiratory specimens, (2) a four-fold or greater increase in titer in paired sera of MP-specific IgG or particle agglutination antibody, or (3) seroconversion of MP-specific IgM in paired sera [3]. Respiratory specimens included nasopharyngeal swab/aspiration, transtracheal aspiration, sputum, and bronchial lavage fluids in certain cases.

### 2.5. PCR and Macrolide Resistance Testing

Nucleic acid amplification tests, including multiplex PCR tests, were conducted at each participating hospital using commercial PCR test kits. For PCR tests used to detect MP in respiratory specimens, the sensitivity generally ranges from 80% to 100%, depending on the specific test and the quality of the sample [18]. The specificity is typically high as well, often exceeding 90%. Macrolide resistance was assessed by identifying point mutations in domain V of the 23S rRNA in the 50S bacterial ribosomal subunit. The mutations included in the analysis were A2063G and A2064G. Macrolide resistance was defined as the presence of either mutation, as these have been shown in experimental studies to confer resistance in vitro [19]. Appendix A describes the brand of PCR kits and the antibody test methods used for diagnosis in each hospital.

### 2.6. Assessment of Disease Severity

The severity of the pneumonia was evaluated using several parameters. Dyspnea at presentation was noted, given its relevance to disease progression. Hospitalization status and the need for respiratory support, including simple oxygen supply, were the primary indicators. Additionally, the length of hospital stay and utilization of intensive care unit resources were key factors in assessing disease severity. Ultimately, patient outcomes, including mortality, were included as significant metrics. While the Pneumonia Severity Index (PSI) or PORT Score is typically used to estimate morbidity and mortality in community-acquired pneumonia, this study adapted relevant parameters from these scores to suit the pediatric population without fully implementing the entire scoring system, due to the specific age group being studied [20].

### 2.7. Statistical Analyses

All statistical analyses were performed using SPSS Statistics for Windows version 28.0 (IBM Corp., Armonk, NY, USA). Categorical data were assessed using the chi-squared test or Fisher’s exact test, as appropriate. Differences between independent groups were evaluated using the Mann–Whitney U test. The significance level was set at *p* < 0.05.

### 2.8. Ethical Considerations

This study received approval from the Institutional Review Boards of CHA Bundang Medical Center (CHAMC2024-01-049-002, 19 February 2024), Samsung Medical Center (2024-01-087-002, 14 February 2024), Seoul St. Mary’s Hospital (KIRB-20240116-018, 19 April 2024), Inha University Hospital (IUH2024-02-020, 20 February 2024), Nowon Eulji Medical Center (EMCS-2024-02-001, 8 February 2024), Seoul National University Children’s Hospital (H-2401-016-1488, 10 January 2024), Korea University Anam Hospital (2024AN0031, 4 December 2023), Seoul National University Bundang Hospital (B-2402-883-401, 15 April 2024), Chungnam National University Hospital (2024-02-004, 14 March 2024), Hallym University Kangnam Sacred Heart Hospital (2024-01-015, 16 February 2024), Pusan National University Yangsan Hospital (55-2024-012, 23 February 2024), Chungbuk National University Hospital (2024-01-005, 25 January 2024), and Jeju National University Hospital (JEJUNUH-2024-01-014, 31 January 2024). The requirement for informed consent from the study participants was waived because of the retrospective nature of the chart review, ensuring patient confidentiality and ethical compliance.

## 3. Results

### 3.1. Epidemiological Profile and Clinical Characteristics

Between September and December 2023, 474 children and adolescents with radiologically confirmed pneumonia and possible evidence of MP infection were screened from 13 hospitals in South Korea. Ultimately, 374 patients were included in the current study as they met at least one of the criteria for microbiological confirmation. Patients excluded from the study primarily included those diagnosed solely based on a positive IgM test, as IgM antibodies can persist for months after an MP infection and may not indicate a current infection.

The median age of the patients was 7.7 (interquartile range (IQR), 5.4–9.6) years, with the majority falling into the 5–9-year-old age group (215, 57.5%) (Table 1). Almost all patients (370, 98.9%) experienced fever, with 66.6% (*n* = 249) having a high fever of 39.0 ℃ or above. The median fever duration was 8 (IQR, 6–10) days. Radiologically, the most prevalent finding was lobular/lobar consolidation (221, 59.1%), followed by peribronchial/parahilar infiltration (76, 20.3%) and patchy consolidation (64, 17.1%). Pleural effusion was observed in 23.3% (*n* = 87) of all patients, with 6.1% (*n* = 23) of all patients having an effusion of 1 cm or larger.

Among the patients included in the study, the majority (331, 88.6%) required hospitalization. Demographic characteristics, including age and sex, did not significantly differ between those treated in outpatient and inpatient settings. However, hospitalized patients were more likely to present with symptoms such as dyspnea (13.9% vs. 2.3%, *p* = 0.027) and findings of pleural effusion (25.7% vs. 4.7%, *p* < 0.001) on chest radiographs. The median length of stay was 5 (IQR, 4–7) days. Percutaneous catheter drainage was performed in 2.1% (*n* = 7) of the admitted patients. Of the hospitalized patients, 1 (0.3%) and 3 (0.9%) of them required invasive mechanical ventilation and intensive care unit admission, respectively. No in-hospital mortalities were reported.

### 3.2. Antibiotic and Corticosteroid Utilization Patterns in MP Treatment

Macrolides were the most commonly used antibiotics, administered to the majority of patients (315, 84.2%), followed by tetracyclines (97, 25.9%) and quinolones (41, 11.0%) (Table 2). Corticosteroids were also frequently used (208, 55.6%). The duration of prescription varied among therapeutic agents, with quinolones (10 days, IQR 7–13) having the longest duration and corticosteroids (5 days, IQR 3–6) the shortest. The interval between fever onset and initiation of medication was significantly shorter for macrolides (4 days, IQR 5–7, *p* < 0.001), while no significant differences were detected for tetracyclines (7 days, IQR 5–9, *p* = 0.737) and quinolones (7 days, IQR 6–10, *p* = 0.145).

A large proportion of the patients were treated with multiple therapeutic agents (Figure 1). Prednisolone was the most common drug combined with macrolides (188, 50.3%). Additionally, prednisolone was co-administered with secondary antibiotics, including tetracyclines and quinolones, in 18.7% (*n* = 70) of the patients. A minority of patients (53, 14.2%) were treated with all three therapeutic agents: macrolides, prednisolone, and secondary medications. Further analysis revealed differences in drug utilization between inpatient and outpatient settings, with overall lower usage of antibiotics and other medications observed in the outpatient group.

Moreover, differences in the age of patients who were prescribed secondary antibiotics were compared. The median age was 9.3 (IQR, 8.0–10.6) years for the tetracycline group and 7.1 (IQR, 5.4–8.2) years for the quinolone group, with the difference achieving statistical significance (*p* < 0.001).

### 3.3. Impact of Macrolide Resistance

In total, 107 patient specimens were processed to identify macrolide resistance at a rate of 87.0% (*n* = 97) (Table 3). All identified mutations associated with macrolide resistance were characterized by the A2063G substitution. Clinical characteristics, including age, sex, symptoms, and radiological findings, were not significantly different between the groups, except for a longer duration of fever in the macrolide-resistant group (median 8 vs. 6 days, *p* = 0.017). Moreover, a notably higher percentage of patients in the macrolide-resistant group were administered tetracycline antibiotics than those in the non-resistant group (35.1% vs. 0%, *p* = 0.005). Although a similar trend was observed for quinolones, the difference was not statistically significant, likely because of the lower overall utilization of quinolones in the macrolide-resistant group.

### 3.4. Influence of Radiologic Findings

Clinical data were analyzed based on radiologic findings (Table 4). Patients exhibiting lobular/lobar consolidation showed a higher proportion of high fever (71.0% vs. 60.1%, *p* = 0.034) and a longer duration of fever (median 8 vs. 7 days, *p* = 0.020) than those with other radiologic findings. Additionally, pleural effusion was more frequently observed in the lobular/lobar consolidation group (30.8% vs. 12.4%, *p* < 0.001), resulting in a higher hospitalization rate (92.3% vs. 83.0%, *p* = 0.008). Conversely, patients without consolidation, including those with other radiologic findings, required respiratory support (including simple oxygen delivery) more frequently (11.8% vs. 5.4%, *p* = 0.009). Furthermore, corticosteroid use was more common in the consolidation group (60.6% vs. 47.1%, *p* = 0.011).

When patients with lobular/lobar consolidation or pleural effusions were grouped together and compared with others, similar trends were observed, including a longer length of hospital stay (median, 6 [IQR 4–7] vs. 5 [IQR 4–6] days; *p* = 0.024).

## 4. Discussion

The prevalence of macrolide resistance observed in this study was high at 87.0%. In our study, we found that macrolides were the most commonly prescribed medication (84.2%). In cases of macrolide-resistant MP infections, secondary medications were more commonly employed, with tetracylcines being preferred over quinolones. Corticosteroids were widely administered, surpassing the use of secondary antibiotics at a frequency of 55.6%. Our study also suggests that macrolide resistance and/or lobular/lobar consolidation in radiological findings may influence disease severity, as indicated by factors such as the duration of fever or hospitalization.

In certain Asian countries, the prevalence of macrolide resistance in MP infections has increased significantly compared to other regions worldwide, with some nations maintaining persistently high levels [5,6,7]. For instance, in Korea, the resistance rate surged from nearly 0% in 2000 to 95.0% during the 2014–2016 epidemic, and has since remained high, reaching up to 80% during the 2019–2020 epidemic [8,15,19]. In contrast, studies investigating macrolide resistance in the United States have reported rates ranging from 3.5% to 13.2% [21]. While the precise reasons for this phenomenon remain unclear, extensive macrolide use appears to be a significant contributing factor. Additionally, the genetic expansion of certain MP strains has been linked to increased resistance rates [8,15]. However, evidence from Japan suggests that strategic macrolide stewardship programs can effectively mitigate resistance rates [22]. A recent study from Korea indicated a notable decrease in macrolide resistance by 66.7%, in contrast to earlier findings where resistance rates remained high [23]. This discrepancy may be attributed to differences in the study population, particularly the focus of the current study solely on hospitalized patients. Interestingly, recent findings from Taiwan reported a significant drop in macrolide resistance, from 85.7% in 2020 to 0% during the 2022–2023 epidemic season, with no substantial genetic differences observed, leaving the reasons for this decline unclear [24]. The mechanism of macrolide resistance observed in this study is solely attributed to the A2063 mutation. It is important to note that several other mechanisms of macrolide resistance exist, including mutations such as C2617G, A2063G/C/T, and A2064G/C [25]. However, the predominant mechanism of macrolide resistance in South Korea has remained unchanged [8,26]. Limited studies conducted in Hong Kong post-pandemic consistently indicate that the major mechanism of macrolide resistance has not undergone significant alterations [27]. This suggests that the A2063G mutation continues to be the primary driver of macrolide resistance both locally and globally.

To the best of our knowledge, few studies have comprehensively examined drug utilization patterns in patients with MP pneumonia [28,29]. Our study observed a high prevalence of macrolide use despite the concurrent high macrolide resistance rate. Notably, doxycycline was preferred over levofloxacin, though the underlying reasons for this preference remain unclear. It is plausible that the liberalization of doxycycline use by the American Academy of Pediatrics, even during the neonatal period, may have influenced this pattern [30]. This is further supported by the recent perspectives that doxycycline should no longer be contraindicated in children [31]. Intriguingly, the median age of patients prescribed levofloxacin was significantly lower than that of those prescribed doxycycline (7.1 vs. 9.3 years, *p* < 0.001), which is inconsistent with the mentioned guidelines. Further research is warranted to better understand these patterns and should be considered in the development of future guidelines for managing intractable or macrolide-resistant MP pneumonia.

Another notable finding is the high rate of β-lactam use, which is interesting given that MP, an atypical bacterium lacking a cell wall, is inherently resistant to β-lactam antibiotics. The use of β-lactams may suggest possible co-infections with β-lactam-susceptible pathogens such as *Streptococcus pneumoniae*. However, considering the widespread use of the pneumococcal conjugate vaccine, the likelihood of such co-infections seems low [32]. This raises questions about the necessity of combined antimicrobial therapy, which warrants further evaluation.

Intriguingly, along with the use of direct antibacterial agents, corticosteroid use was prevalent in our study cohort, with a significant number of patients concurrently receiving both macrolides and corticosteroids. With a macrolide resistance rate of nearly 90% in our study, the therapeutic efficacy of macrolides appears questionable, despite frequent reports of their anti-inflammatory effects [33]. However, the discrepancy between in vivo and in vitro impacts of macrolide resistance suggests that macrolides might still demonstrate efficacy in managing macrolide-resistant MP infections, although depending solely on macrolide therapy may prove insufficient. Moreover, the self-limiting nature of non-severe MP infections complicates the assessment of macrolide efficacy. Physicians may have chosen corticosteroids over secondary antibiotics based on clinical judgment [34].

Alternatively, patients treated solely with macrolides or corticosteroids may not have required additional antibiotic therapy, raising questions about the necessity of additional treatment in some cases [3]. For instance, a recent study from China showed similar clinical outcomes with macrolides, regardless of macrolide resistance in MP pneumonia [35]. Considering the aim of this study was to provide an overview of the current MP pneumonia outbreak, and given the lengthiness and complexity of additional investigations, we had to focus this manuscript on the core findings. To address the more detailed questions regarding treatment efficacy and utilization patterns, we have decided to conduct a new and separate study. This upcoming research will utilize treatment data from the current and expanded dataset to delve deeper into the efficacy of various treatment modalities on fever duration in Korean children with MP pneumonia, integrating the timeline of multiple medications. This follow-up study aims to provide a more comprehensive understanding and address the questions that remain unanswered in the current manuscript.

Debates persist regarding the clinical implications of macrolide resistance in MP infections, particularly with respect to disease severity and progression. Although it is generally acknowledged that macrolide-resistant MP infections tend to be more severe, the evidence remains inconclusive, with studies yielding mixed results [9,21]. In our current study, we were able to assess macrolide resistance in approximately 30% of cases, limiting our ability to draw direct comparisons based on resistance status alone. Although we observed a significantly longer duration of fever and a higher proportion of patients experiencing dyspnea in the resistant group, the macrolide-sensitive group exhibited a higher prevalence of high fever. Although some reports have suggested a higher incidence of lobar pneumonia in the resistant group, our findings did not reveal significant differences in radiologic findings between the two groups [36]. Furthermore, the length of hospital stay and clinical outcomes were similar in both groups.

Notably, the prescription rates of secondary antibiotics and corticosteroids were significantly higher in the macrolide-resistant group. However, it remains uncertain whether these differences in drug utilization patterns directly influence clinical outcomes, as they may be influenced by the results of macrolide resistance testing. Interestingly, no patients in the macrolide-sensitive group received secondary antibiotics. While this may suggest that physicians were aware of macrolide sensitivity and tailored their treatment accordingly, the lower utilization of corticosteroids in this group indicates a multifactorial approach to drug selection based on resistance status.

Some studies have highlighted the importance of considering clinical findings alongside radiological presentations, rather than solely relying on the macrolide resistance status, to predict disease outcomes [37]. Our findings are consistent with those of such studies. Patients with lobular/lobar consolidation, with or without pleural effusion, experienced prolonged fever and were hospitalized more frequently. However, given the similar outcomes of MP pneumonia, hospitalization may have been influenced more by the attending physicians’ concerns than by clinical manifestations alone. The higher frequency of respiratory support, including oxygen delivery, in patients without pleural effusion or consolidation is particularly noteworthy. This suggests that partial consolidation alone may not lead to hypoxemia; instead, diffuse infiltration may have a more significant effect on the patient’s condition. Furthermore, we observed increased corticosteroid use in the lobular/lobar consolidation group, indicating that physicians may opt for corticosteroids more frequently in the presence of consolidation.

While our multi-site approach provided a comprehensive perspective on MP outbreaks, several limitations should be acknowledged. First, variations in drug utilization patterns and admission criteria across different study sites may have introduced inconsistencies in our data. Additionally, since the study was conducted during an ongoing epidemic, the dynamic nature of the outbreak may not have been fully captured, potentially affecting the generalizability of our findings. A fundamental limitation was the restricted availability of tests for macrolide resistance, which limited our ability to perform comprehensive comparisons based on resistance status. It is also likely that there were differences in the indications for testing macrolide resistance among and within institutions, influenced by the testing capacity. For example, patients with more intractable cases were more likely to be tested for macrolide resistance, potentially skewing our resistance rates. Furthermore, the fact that this study was conducted in academic university hospitals, where more severe cases are often treated, may have also influenced the high resistance rates observed. As such, the macrolide resistance data presented in this study should be interpreted with caution. Future studies utilizing more sophisticated surveys to explore drug utilization patterns in MP infections, particularly in relation to macrolide resistance, are needed to provide more detailed insights. Additionally, the planned investigation into corticosteroid dosages could not be thoroughly pursued due to insufficient data, a topic that needs further exploration given emerging evidence supporting higher doses [34,38].

Despite these limitations, our study is the first investigation of an MP outbreak in Korea following the COVID-19 pandemic and provides a comprehensive examination of this epidemic. Our findings have the potential to inform clinical practice beyond the scope of general hospitals by evaluating drug utilization patterns and therapeutic effects in tertiary hospitals. The high prevalence of macrolide resistance, along with multidrug usage patterns, underscores the complexity of MP infection management. Furthermore, the widespread use of corticosteroids and the preference for doxycycline over levofloxacin underscore the importance of carefully considering treatment strategies in the context of evolving resistance patterns. However, we cannot provide a definitive recommendation for patients unresponsive to MP-based treatment, suggesting the need to consider alternative causes in such cases where MP may not be the primary factor.

## 5. Conclusions

In summary, our study highlights the significant challenges posed by macrolide resistance in MP outbreaks following the COVID-19 pandemic, emphasizing the need for tailored therapeutic approaches. Despite a high macrolide resistance rate of 87.0%, macrolides remained the most commonly prescribed antibiotic with no fatal outcomes, prompting questions about the optimal usage of alternative treatments. Our findings reveal a complex interplay between macrolide resistance, drug utilization patterns, and clinical outcomes, with frequent use of corticosteroids alongside macrolides. These insights underscore the importance of developing more strategic and evidence-based guidelines.

## Figures and Tables

**Figure 1 microorganisms-12-01806-f001:**
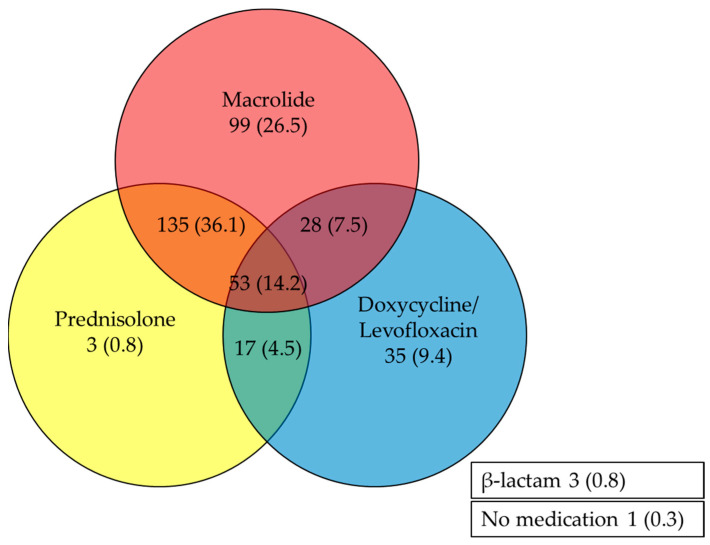
Numbers and percentages of antibiotics and corticosteroid utilization in managing *Mycoplasma pneumoniae* pneumonia.

**Table 1 microorganisms-12-01806-t001:** Clinical and laboratory characteristics and management of children and adolescents with *Mycoplasma pneumoniae* pneumonia.

Characteristic	Total (*n* = 374)
Age (median, IQR)	7.7 (5.4–9.6)
<2 y	14 (3.7)
2–4 y	64 (17.1)
5–9 y	215 (57.5)
10–18 y	81 (21.7)
Male	203 (54.3)
Symptoms	
Fever	370 (98.9)
High fever (≥ 39.0 °C)	249 (66.6)
Duration (median, IQR)	8 (6–10)
Dyspnea	47 (12.6)
Radiologic findings	
Lobular/lobar consolidation	221 (59.1)
Peribronchial/parahilar infiltration	76 (20.3)
Patchy consolidation	64 (17.1)
Nodules	6 (1.6)
Total haziness (one or both lung)	4 (1.1)
GGO (one or both lung)	3 (0.8)
Other radiologic findings	
Pleural effusion	87 (23.3)
Small (<1 cm)	64 (17.1)
Large (≥1 cm)	23 (6.1)
Atelectasis	3 (0.8)
CRP (mg/dL) (median, IQR)	2.27 (0.98–4.66)
ESR (mm/h) (median, IQR)	34 (22–47)
PCT (ng/mL) (median, IQR)	0.11 (0.08–0.22)
LDH (U/L) (median, IQR)	357 (287–465)
Inpatient	331 (88.6)
Length of stay (days) (median, IQR)	5, 4–7
Respiratory support	29/331 (8.8)
Percutaneous catheter drainage	7/331 (2.1)
Intensive care unit admission	3/331 (0.9)
Invasive mechanical ventilation	1/331 (0.3)
Death in the hospital	0/331 (0)
Macrolide resistance	93/107 (87.0)

IQR, interquartile range; GGO, ground-glass opacity; CRP, C-reactive protein; ESR, erythrocyte sedimentation rate; PCT, procalcitonin; LDH, lactate dehydrogenase. Variables without specific indications are represented as No. (%). Respiratory support includes minimal oxygen supply to the application of mechanical ventilators. All macrolide resistance cases were attributed to the A2063G mutation.

**Table 2 microorganisms-12-01806-t002:** Therapeutic management of children and adolescents with *Mycoplasma pneumoniae* pneumonia.

Characteristic	Medication Prescribed	Total Length of Prescription	Fever to Medication
	No. (%) (*n* = 374)	Median, IQR (Days)	
Therapeutic agent			
β-lactam antibiotics	211 (56.4)	5, 4–8	
Macrolide antibiotics	315 (84.2) *	7, 4–11 *	4, 5–7 *
Tetracycline antibiotics	97 (25.9) *	9, 7–11 *	7, 5–9
Quinolone antibiotics	41 (11.0) *	10, 7–13 *	7, 6–10
Corticosteroids	208 (55.6)	5, 3–6	7, 5–9

IQR, interquartile range. Asterisks indicate a significance of *p* < 0.05 compared to corticosteroid treatment. All tetracycline antibiotics were doxycycline, while quinolone antibiotics consisted of levofloxacin (32, 78.0%) and moxifloxacin (9, 22.0%).

**Table 3 microorganisms-12-01806-t003:** Comparison of clinical characteristics, treatment strategies, and outcomes in *Mycoplasma pneumoniae* pneumonia based on macrolide resistance status.

Characteristics	Macrolide Resistance		*p*-Value
	Susceptible (*n* = 14)	Resistant (*n* = 97)	
Age (median, IQR)	8.3 (4.0–9.6)	7.6 (5.4–9.2)	0.543
Male	9 (64.3)	62 (63.9)	0.979
Symptoms			
Fever	13 (92.9)	96 (99.0)	0.237
High fever (≥39.0 °C)	9 (64.3)	38 (39.2)	0.075
Duration (median, IQR)	6 (5–7)	8 (7–11)	0.017
Dyspnea	2 (14.3)	19 (19.6)	>0.999
Radiologic findings			
Lobular/lobar consolidation	10 (71.4)	67 (69.1)	>0.999
Peribronchial/parahilar infiltration	2 (14.3)	18 (18.6)	>0.999
Patchy consolidation	2 (14.3)	10 (10.3)	>0.999
Nodules	0	0	N/A
Total haziness (one or both lung)	0	2 (2.1)	>0.999
GGO (one or both lung)	0	0	N/A
Other radiologic findings			
Pleural effusion	6 (42.9)	25 (25.8)	0.209
Inpatient	12 (85.7)	84 (86.6)	>0.999
Length of stay (days) (median, IQR)	5.5 (5–6.5)	6 (4–7.5)	0.804
Respiratory support	0	11/84 (26.2)	0.214
Percutaneous catheter drainage	0	2/84 (2.4)	>0.999
Intensive care unit admission	0	3/84 (3.6)	>0.999
Invasive mechanical ventilation	0	1/84 (1.2)	>0.999
Therapeutic agent			
β-lactam antibiotics (IV and/or oral)	11 (78.6)	73 (75.3)	>0.999
Macrolide antibiotics	10 (71.4)	78 (80.4)	0.483
Tetracycline antibiotics	0	34 (35.1)	0.005
Quinolone antibiotics	0	20 (20.6)	0.070
Corticosteroid	3 (21.4)	44 (45.4)	0.146

MP, *Mycoplasma pneumoniae;* IQR, interquartile range; GGO, ground-glass opacity; IV, intravenous. Variables without specific indications are represented as No. (%).

**Table 4 microorganisms-12-01806-t004:** Comparison of clinical characteristics, management, and outcome of *Mycoplasma pneumoniae* pneumonia according to radiologic findings.

Characteristics	Radiologic Findings		*p*-Value
	Lobular/Lobar Consolidation (*n* = 221)	Others (*n* = 153)	
Age (median, IQR)	7.9 (5.8–9.8)	7.3 (4.8–9.7)	0.187
Male	121 (54.8)	82 (53.6)	0.834
Symptoms			
Fever	219 (99.1)	151 (98.7)	>0.999
High fever (≥39.0 °C)	157 (71.0)	92 (60.1)	0.034
Duration (median, IQR)	8 (6–11)	7 (6–9)	0.020
Dyspnea	25 (11.3)	22 (14.4)	0.429
Other radiologic findings			
Pleural effusion	68 (30.8)	19 (12.4)	<0.001
CRP (mg/dL) (median, IQR)	2.48 (1.01–5.17)	2.06 (0.94–3.99)	0.247
ESR (mm/h) (median, IQR)	35 (23–52)	30.5 (20–43)	0.404
PCT (ng/mL) (median, IQR)	0.12 (0.08–0.23)	0.11 (0.07–0.22)	0.537
LDH (U/L) (median, IQR)	377 (287–476)	350 (286–437)	0.164
Inpatient	204 (92.3)	127 (83.0)	0.008
Length of stay (days) (median, IQR)	5 (4–6)	5 (4–6)	0.228
Respiratory support	11/204 (5.4)	18/153 (11.8)	0.009
Percutaneous catheter drainage	5/204 (2.5)	2/153 (1.3)	0.703
Intensive care unit admission	0	3/153 (2.0)	0.078
Invasive mechanical ventilation	0	1/153 (0.7)	0.429
Therapeutic agent			
β-lactam antibiotics (IV and/or oral)	119 (53.8)	92 (60.1)	0.244
Macrolide antibiotics	182 (82.4)	133 (86.9)	0.251
Tetracycline antibiotics	60 (27.1)	37 (24.2)	0.550
Quinolone antibiotics	29 (13.1)	12 (7.8)	0.130
Corticosteroid	134 (60.6)	72 (47.1)	0.011

IQR, interquartile range; CRP, C-reactive protein; ESR, erythrocyte sedimentation rate; PCT, procalcitonin; LDH, lactate dehydrogenase. Variables without specific indications are represented as No. (%).

## Data Availability

The raw data supporting the conclusions of this article will be made available by the authors on request.

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
