# Peer review of "Clinical Manifestations, Macrolide Resistance, and Treatment Utilization Trends of Mycoplasma pneumoniae Pneumonia in Children and Adolescents in South Korea"

_microorganisms, 2024, doi:10.3390/microorganisms12091806_

Round 1

Reviewer 1 Report

Comments and Suggestions for Authors

A highly relevant paper which should be shared with the pediatric community. The paper is well written, and the research purposes well presented. I do not have any major concerns. 
Few minor issues: how was the resistance defined? What would you rather suggest physicians when a patient does not respond to a MP-based treatment. Moreover, please extent the limitation section and cite the relevant MP-based literature which were recently published (2023/2024).

Comments on the Quality of English Language

Good.

Author Response

Reviewer 1:

A highly relevant paper which should be shared with the pediatric community. The paper is well written, and the research purposes well presented. I do not have any major concerns.

Response: We sincerely appreciate the reviewer’s positive feedback and recognition of our study’s relevance. We have carefully addressed the reviewer’s suggestions and made the necessary corrections and improvements to the manuscript.

Few minor issues:

How was the resistance defined?

Response: Macrolide resistance is defined by the presence of point mutations in domain V of the 23S rRNA gene, specifically at positions A2063, A2064, and A2067. These mutations are well-documented in experimental studies, which demonstrate a strong correlation with macrolide resistance in vitro. Based on the reviewer’s suggestion, we have included an additional reference to further illustrate the mechanisms and their implications for resistance.

What would you rather suggest physicians when a patient does not respond to a MP-based treatment.

Response: Thank you for raising this important issue. Unfortunately, there isn’t a definitive answer. One approach is to escalate to secondary antibiotics, or as discussed in this study, corticosteroids may be a viable alternative. However, it’s also crucial to consider the possibility of MP colonization rather than infection, or coinfections with other pathogens. In such cases, alternative diagnoses should be explored. We have added this consideration to the limitations section.

Moreover, please extent the limitation section and cite the relevant MP-based literature which were recently published (2023/2024).

Response: Thank you for your valuable comment. We have revised the manuscript to include the most recent and relevant literature from 2023 and 2024, with appropriate descriptions added where necessary. The following references have been included:

Li Y, Wu M, Liang Y, et al. Mycoplasma pneumoniae infection outbreak in Guangzhou, China after COVID-19 pandemic. Virol J. 2024;21(1):183. doi:10.1186/s12985-024-02458-z

Jiang M, Zhang H, Yao F, et al. Influence of non-pharmaceutical interventions on epidemiological characteristics of Mycoplasma pneumoniae infection in children during and after the COVID-19 epidemic in Ningbo, China. Front Microbiol. 2024;15:1405710. doi:10.3389/fmicb.2024.1405710

He M, Xie J, Rui P, et al. Clinical efficacy of macrolide antibiotics in Mycoplasma pneumoniae pneumonia carrying a macrolide-resistant mutation in the 23S rRNA gene in pediatric patients. BMC Infect Dis. 2024;24(1):758. doi:10.1186/s12879-024-09612-6

Wu TH, Fang YP, Liu FC, et al. Macrolide-Resistant Mycoplasma pneumoniae Infections among Children before and during COVID-19 Pandemic, Taiwan, 2017-2023. Emerg Infect Dis. 2024;30(8):1692-1696. doi:10.3201/eid3008.231596

Reviewer 2 Report

Comments and Suggestions for Authors

The present paper gives an excellent insight into the clinical practices regarding the management of Mycoplasma pneumoniae infections, especially in those patients who were hospitalized due to a more difficult course.

The findings contribute significantly to the understanding of the course and outcome of Mycoplasma pneumonia infection in children and adolescents caused by macrolide-resistant Mycoplasma pneumoniae.

Perhaps you should add exact starting and ending dates of the study.  

I have no other comments or suggestions for the improvement.

Congratulations for good work done.

Author Response

Reviewer 2:

The present paper gives an excellent insight into the clinical practices regarding the management of Mycoplasma pneumoniae infections, especially in those patients who were hospitalized due to a more difficult course.

The findings contribute significantly to the understanding of the course and outcome of Mycoplasma pneumonia infection in children and adolescents caused by macrolide-resistant Mycoplasma pneumoniae.

Perhaps you should add exact starting and ending dates of the study. 

I have no other comments or suggestions for the improvement.

Congratulations for good work done.

Response: Thank you very much for your encouraging comments and insightful feedback. We are pleased to hear that our work provides valuable insights into the management of Mycoplasma pneumoniae infections. Based on your suggestion, we have included the exact starting and ending dates of our study period to enhance the clarity of our research timeframe. We appreciate your positive evaluation and constructive suggestion, which have greatly enriched our manuscript.

Reviewer 3 Report

Comments and Suggestions for Authors

This is an excellent well-written study evaluating pediatric patients with pneumonia caused by Mycoplasma pneumoniae and the impact of macrolide resistance and different radiological findings on patients outcomes. A few issues need to be addressed as outlined below:

1. Abstract (line 40): Change "enrolled" to "evaluated" or "screened" as the word "enrolled" is usually used in prospective studies. Same on line 174 of the results.

2. Section 2.6. Assessment of disease severity: These are valid metrics, but why didn't the authors simply utilize validated scores, such pneumonia severity index (PSI)? Please provide a justification at the end of this section or in the discussion.

3. Section 3.2. Antibiotic and Corticosteroid Utilization Patterns in MP Treatment; Did you record the doses used? Were macrolides used at higher than standard doses in cases exhibiting macrolide resistance? If that piece of information wasn't captured, please include it as a limitation since clinicians typically tend to use higher than standard doses in cases of intermediate or full resistance to an antibiotic.

4. Lines 230 and 272: Please remove "(data not shown)."

5. Line 272: It would be helpful to add the IQR since the difference between the two medians is only one day; thus, may not be considered clinically significant despite being statistically significant.

6. Figure 1: This is an excellent figure, but please put "β-lactam" and "No medication" each in an enclosed circle or a rectangle as they currently appear as a free text not belonging to the figure.

7. Discussion: Please mention the % sensitivity and specificity of PCR tests used to detect MP in respiratory specimens. This could be added somewhere in the 2nd paragraph or wherever you see fit.

8. Discussion (3rd paragraph): Please comment on the use of β-lactam antibiotics which aren't active against Mycoplasma pneumoniae which belongs to the group of atypical bacteria lacking cell walls.

9. Discussion (3rd paragraph): Regarding the use of doxycycline in pediatric patients, please include in your argument the new perspective by McCreary, et al (find it attached; https://doi.org/10.1093/cid/ciad357) that doxycycline should no longer be contraindicated in children in this article (page 1123).

Author Response

Reviewer 3:

This is an excellent well-written study evaluating pediatric patients with pneumonia caused by Mycoplasma pneumoniae and the impact of macrolide resistance and different radiological findings on patients’ outcomes. A few issues need to be addressed as outlined below:

Response: Thank you very much for your positive feedback and constructive suggestions. We are delighted to hear that our study on pediatric Mycoplasma pneumoniae infections has provided valuable insights. We are addressing the issues you outlined to ensure the manuscript is as thorough and clear as possible. We look forward to enhancing the paper with your recommendations and appreciate your support in refining our work.

  1. Abstract (line 40): Change "enrolled" to "evaluated" or "screened" as the word "enrolled" is usually used in prospective studies. Same on line 174 of the results.

Response: Thank you for your insightful observations. We have replaced the term "enrolled" with "screened" to more accurately reflect the retrospective nature of our study. This change has been applied consistently throughout the manuscript, including in the abstract and results section, to ensure clarity and precision in our terminology.

  1. Section 2.6. Assessment of disease severity: These are valid metrics, but why didn't the authors simply utilize validated scores, such pneumonia severity index (PSI)? Please provide a justification at the end of this section or in the discussion.

Response: Thank you for your valuable comment. In our review of the pneumonia severity indices like the PSI/PORT score, we found that these are generally designed for adult populations, which influenced our decision against their direct application for pediatric patients. We have, however, incorporated aspects of these scores suitable for children into our methodology. To clarify this adaptation and address your feedback, we've detailed our approach in the discussion section of our manuscript and included a relevant reference on PSI/PORT scores.

Fine MJ, Auble TE, Yealy DM, et al. A prediction rule to identify low-risk patients with community-acquired pneumonia. N Engl J Med. 1997;336(4):243-250. doi:10.1056/NEJM199701233360402

  1. Section 3.2. Antibiotic and Corticosteroid Utilization Patterns in MP Treatment; Did you record the doses used? Were macrolides used at higher than standard doses in cases exhibiting macrolide resistance? If that piece of information wasn't captured, please include it as a limitation since clinicians typically tend to use higher than standard doses in cases of intermediate or full resistance to an antibiotic.

Response: Thank you for the constructive comment. We recognize the importance of capturing dosage information, especially in relation to macrolide resistance. While we aimed to collect dosage data based on patient body weight, this was challenging due to many patients receiving prescriptions from local clinics, where exact dosage information was unavailable. Additionally, the multicenter nature of the study complicated dosage evaluation further. We acknowledge recent evidence supporting higher doses of corticosteroids and have included this as a limitation in our study, with relevant information added.

Sun LL, Ye C, Zhou YL, Zuo SR, Deng ZZ, Wang CJ. Meta-analysis of the Clinical Efficacy and Safety of High- and Low-dose Methylprednisolone in the Treatment of Children With Severe Mycoplasma Pneumoniae Pneumonia. Pediatr Infect Dis J. 2020;39(3):177-183. doi:10.1097/INF.0000000000002529

Okumura T, Kawada JI, Tanaka M, et al. Comparison of high-dose and low-dose corticosteroid therapy for refractory Mycoplasma pneumoniae pneumonia in children. J Infect Chemother. 2019;25(5):346-350. doi:10.1016/j.jiac.2019.01.003

  1. Lines 230 and 272: Please remove "(data not shown)."

Response: Thank you for your recommendation. The specified corrections have been made to the manuscript.

  1. Line 272: It would be helpful to add the IQR since the difference between the two medians is only one day; thus, may not be considered clinically significant despite being statistically significant.

Response: We appreciate your suggestion and agree with the importance of providing a complete statistical context. Interquartile ranges have been added to the data in question to enhance the manuscript's clarity and relevance of the findings.

  1. Figure 1: This is an excellent figure, but please put "β-lactam" and "No medication" each in an enclosed circle or a rectangle as they currently appear as a free text not belonging to the figure.

Response: Thank you for your valuable feedback. We have recognized the issues with the layout of Figure 1 and have adjusted it by enclosing 'β-lactam' and 'No medication' within rectangles to ensure they are clearly part of the figure. This modification enhances the presentation and readability of the data.

  1. Discussion: Please mention the % sensitivity and specificity of PCR tests used to detect MP in respiratory specimens. This could be added somewhere in the 2nd paragraph or wherever you see fit.

Response: Thank you for the suggestion. We have added the sensitivity and specificity percentages of the PCR tests used to detect Mycoplasma pneumoniae in respiratory specimens in the Methods section. Specifically, PCR tests for M. pneumoniae have a sensitivity ranging from 80% to 100% and a specificity typically exceeding 90%, ensuring reliable detection in clinical settings. This information has been referenced accordingly.

Waites KB, Xiao L, Paralanov V, Viscardi RM, Glass JI. Molecular methods for the detection of Mycoplasma and ureaplasma infections in humans: a paper from the 2011 William Beaumont Hospital Symposium on molecular pathology. J Mol Diagn. 2012;14(5):437-450. doi:10.1016/j.jmoldx.2012.06.001

  1. Discussion (3rd paragraph): Please comment on the use of β-lactam antibiotics which aren't active against Mycoplasma pneumoniae which belongs to the group of atypical bacteria lacking cell walls.

Response: We have added a new paragraph in the Discussion section to address the use of β-lactam antibiotics, explaining that Mycoplasma pneumoniae lacks a cell wall and is thus inherently resistant to these antibiotics. Additionally, we have included a reference that discusses the decreased incidence of invasive bacterial infections by Streptococcus pneumoniae following widespread pneumococcal conjugate vaccination, which further supports the low likelihood of co-infections and raises questions about the necessity of combined antimicrobial therapy.

Song SH, Lee H, Lee HJ, et al. Twenty-Five Year Trend Change in the Etiology of Pediatric Invasive Bacterial Infections in Korea, 1996-2020. J Korean Med Sci. 2023;38(16):e127. Published 2023 Apr 24. doi:10.3346/jkms.2023.38.e127

  1. Discussion (3rd paragraph): Regarding the use of doxycycline in pediatric patients, please include in your argument the new perspective by McCreary, et al (find it attached; https://doi.org/10.1093/cid/ciad357) that doxycycline should no longer be contraindicated in children in this article (page 1123).

Response: Thank you for sharing the invaluable article by McCreary et al. We have added the reference to the manuscript and cited it accordingly. The discussion now includes the new perspective that doxycycline should no longer be contraindicated in children, as detailed in McCreary et al.'s article

McCreary EK, Johnson MD, Jones TM, et al. Antibiotic Myths for the Infectious Diseases Clinician [published correction appears in Clin Infect Dis. 2024 Jun 14;78(6):1780. doi: 10.1093/cid/ciae161]. Clin Infect Dis. 2023;77(8):1120-1125. doi:10.1093/cid/ciad357

Round 2

Reviewer 3 Report

Comments and Suggestions for Authors

Many thanks to the authors for promptly addressing the comments in a timely and professional manner. I endorse the acceptance of the manuscript for publication.